# A Lightweight Authentication Scheme for a Network of Unmanned Aerial Vehicles (UAVs) by Using Physical Unclonable Functions

**Mohammed Saeed Alkatheiri** [1], **Sajid Saleem** [2], **Mohammed Ali Alqarni** [3], **Ahmad O. Aseeri** [4,*], **Sajjad Hussain Chauhdary** [5] and **Yu Zhuang** [6]

1   Department of Cybersecurity, College of Computer Science and Engineering, University of Jeddah, Jeddah 21959 , Saudi Arabia
2   Department of Computer and Network Engineering, College of Computer Science and Engineering, University of Jeddah, Jeddah 21959, Saudi Arabia
3   Department of Software Engineering, College of Computer Science and Engineering, University of Jeddah, Jeddah 21959, Saudi Arabia
4   Department of Computer Science, College of Computer Engineering and Sciences, Prince Sattam Bin Abdulaziz University, Al-Kharj 11942, Saudi Arabia
5   Department of Computer Science and Artificial Intelligence, College of Computer Science and Engineering, University of Jeddah, Jeddah 21959, Saudi Arabia
6   Department of Computer Science, Texas Tech University, Lubbock, TX 79409, USA
*   Correspondence: a.aseeri@psau.edu.sa

**Abstract:** A network of agents constituted of multiple unmanned aerial vehicles (UAVs) is emerging as a promising technology with myriad applications in the military, public, and civil domains. UAVs' power, memory, and size constraints, ultra-mobile nature, and non-trusted operational environments make them susceptible to various attacks, including physical capturing and cloning attacks. A robust and resilient security protocol should be lightweight and resource-efficient in addition to providing protection against physical and tampering threats. This paper proposes an authentication protocol for a UAV-based multi-agent system robust against various threats and adversaries, including strong resistance against cloning and physical attacks. The proposed protocol is based on a physical unclonable function (PUF), a well-known hardware security primitive that is utilized for low-cost authentication and cryptographic key generation. The analysis of the proposed approach shows that it provides strong protection against various attacks, including tampering and cloning, and exhibits scalability and energy efficiency.

**Keywords:** IoT security; authentication; physical unclonable functions; unmanned aerial vehicles; tampering attack

## 1. Introduction

Recent advances in the computing, communication, control, and fabrication of miniaturized electronics along with some regulatory relaxations have propelled unmanned aerial vehicles (UAVs) to the forefront of emerging and promising technologies [1,2]. Unmanned aerial vehicles (UAVs) are considered to have an immense potential in diverse applications representing the public, civil, and military domains. Specifically, UAVs hold great promise for surveillance, reconnaissance, patrolling, transportation management, disaster response and recovery, inventory management, etc. [3]. UAVs have become more widespread than ever due to rapid technological advancements in their capabilities and cost reductions, making them a promising solution of tremendous potential for future applications.

The most critical and exclusive applications involve missions where UAVs can be swiftly deployed to provide services in a hazardous or disaster-affected area. The physical deployment of personnel in such a situation might involve huge unwarranted risks to

human lives and critical assets. A system with multiple UAVs, or a network of drones, can enable the execution of collaborative strategies and coordinated formations typically required for planning and accomplishing complicated tasks and missions. However, despite their numerous merits, multi-agent UAV networks introduce complexities in the design of protocols and are susceptible to multiple vulnerabilities and attacks. Existing routing and security solutions developed for ad hoc mesh networks and the Internet of things (IoT) are unsuitable due to the peculiar characteristics and constraints of multi-agent UAV networks [1].

Security and privacy are critical concerns for UAV-based aerial networks in most civilian- and military-based applications due to their ultra-mobile nature, distributed deployment, and use of the wireless medium for communication. These factors make them prone to physical attacks and susceptible to cyberthreats. These attacks on UAV-based networks include physical tampering, cloning, and capturing attacks, replay attacks, man-in-the-middle attacks, etc. In a scenario involving multiple UAVs deployed for monitoring and surveillance of critical military or strategic infrastructure, an adversarial drone can deceive the network into believing that it is a legitimate node of the network if it can successfully authenticate itself onto the network. Such a malicious node can launch various attacks on the network and its devices. Moreover, it can easily eavesdrop on sensitive data and control signals, and capture or impersonate legitimate UAV nodes that constitute the network. Such an adversarial action can severely compromise the strategic assets, likely leading to a tremendous loss of human lives and infrastructure.

Authentication of individual UAVs and the ground station is a critical security requirement for safe execution of all tasks and missions that might involve the transfer of sensitive data, commands, and mission parameters among nodes as well as with central command and the control module. Authentication is typically accomplished through some shared secret between two communicating agents. Prevalent authentication techniques designed for the IoT and mobile ad hoc networks (MANETs) typically utilize a key store for shared secrets that enable cryptographic operations [4–6]. However, these approaches are unsuitable for UAV-based aerial networks for various reasons. First, the nodes can be captured and subsequently controlled by the adversaries, thus leading to the failure of the overall secure operation of the network. Secondly, most reliable security protocols involve complicated cryptographic techniques that are computationally challenging and conspicuously consume an already limited power source available to a UAV node. Even though some recent works address the topic of secure authentication and privacy for a network of UAVs, most of these approaches are preliminary and there exists significant scope for improvement with respect to computational and power efficiency, decentralization of the network, and reduction of delay caused by security protocols.

In this paper, we propose a secure lightweight authentication mechanism for a multi-UAV system. Earlier approaches exist for authentication of devices on a UAV network, but they fail to provide physical security as the secret keys are stored on the memory of a device. Storing the cryptographic keys on device memory exposes the UAV nodes and the network to several physical and side-channel attacks. The objective of the proposed authentication technique is to address this issue in the existing approaches by not storing the cryptographic secrets on device memory. A high level of security is still guaranteed while achieving efficiency of resource utilization and resilience against node capturing and tampering attacks. Below is the list of some significant contributions of this work.

1. A novel secure protocol is proposed that can be used to mutually authenticate a UAV and the ground station. Moreover, it can be used to establish a secure session between the UAV drones and the ground station.
2. The protocol can be modified to establish a secure communication session between any two devices on the network of UAVs.
3. A formal security proof is provided to establish that the proposed protocol is safe and secure for the network of UAVs.

4.　　A performance comparison between state-of-the-art protocols and the proposed protocols shows that the latter is resource- and power-efficient, and simultaneously achieves the objectives of securing the network nodes and its communication.

The rest of the paper is organized as follows. In Section 2, we discuss related work on authentication techniques. Section 3 presents preliminary background about PUFs and UAVs, and Section 4 describes the assumed system model and the attack model. The proposed appraoch is discussed in Section 5. In Section 6, we perform security analysis of the proposed approach, and Section 7 presents the performance analysis of the proposed authentication approach. Conclusions are drawn in Section 8.

## 2. Related Work

With the ever-increasing relevance of UAV-based aerial networks, issues that relate to the security and privacy of these networks are beginning to capture attention of researchers, developers, and users. In the last two decades, we have witnessed plenty of progress in the development of security protocols of mobile ad hoc networks (MANETs), vehicular ad hoc networks (VANETs), and more recently, for the IoT; however, these techniques are not directly applicable to the UAV-based aerial networks [7]. Although some aspects of the UAV networks resemble those of MANETs, VANETs, and IoT, the increased mobility, stricter constraints on resources, and exposure to a greater variety of threats and operational hazard make them more vulnerable as compared to MANETs, VANETs, and IoT [8].

Significant issues in communication and security protocols of UAV-based networks have been identified and reported in [1,9]. The authors in [9] demonstrate that some seemingly complicated attacks, such as GPS spoofing and wifi attacks, can be very easily launched against multi-UAV networks. Their work concludes that security and privacy approaches that are found effective in traditional delay-tolerant networks, e.g., anomaly detection techniques, are entirely unsuitable for UAV networks due to the strict latency, power, memory, and operational constraints. The authors in [1] also identify several security pitfalls in the operation of UAV networks, and stress the need for security authentication, privacy, as well as the requirements of device and data protection. Thus, UAV-based networks must be secured by protocols that can ensure a suitable tradeoff between performance and security. However, these works, i.e., [1,9], identify security issues but do not propose any new solution to efficiently resolve them.

The authors of [10] demonstrate the vulnerability of wifi-based UAVs, specifically the Parrot Bebop UAV, to the address resolution protocol (ARP) cache poisoning, buffer overflow, and basic denial-of-service attacks. These attacks can be launched during the address resolution discovery and connection process, and can lead to catastrophic consequences for the UAV and the network; e.g., a UAV can be stopped inflight or forced to land by the attacker. Authors also present a multi-layer security approach to counter and secure the UAVs against such attacks. Security measures at the physcial layer in a UAV-aided cellular network is also an area of recent interest among the researchers. For example, authors of [11] propose a secure transmission strategy in a system that uses a combination of UAVs with non-orthogonal multiple access (NOMA) techniques to achieve simultaneous wireless information and power transfer (SWIPT). Similarly, the authors of [12] discuss the security issues in UAV networks that use NOMA transmission techniques. Through joint optimization of the hovering position of UAV and power allocation strategy, secure transmission is made possible for the cellular user served by the UAV base station.

A secure transmission scheme is proposed by the authors of [13]. The proposed scheme is shown to be robust against a strong adversarial model and is based upon a continuous renewal of cryptographic keys between the UAV and the ground station. The proposed scheme claims to exhibit secrecy against physical attacks, side-channel attacks and fault-injection attacks. However, a single initial key stored in non-volatile memory is used to generate fresh keys through a hash function. Even though the authors propose using multiple streams of key in parallel, a strong adversary could track these streams if they get hold of any single key.

The authors of [14] propose a scheme for mutual authentication and direct anonymous attestation, abbreviated as MA-DAA based upon the generation of asymmetric pairings and the merging of the identities of the host and the trusted platform module (TPM). The proposed scheme achieves randomization of credentials, batch verification and proof, and mutual authentication. Although the proposed scheme addresses the computation resource challenge of the UAV system, its application is restricted to the devices that support the TPM security co-processors, which are both specialized and expensive.

An efficient low-overhead security protocol for a UAV network operating in a disaster-affected area has been proposed by [15]. The efficiency of the scheme is based upon the lightweight ring-learning with errors (Ring-LWE) encryption scheme. However, the proposed scheme guarantees only one-sided authentication.

A certificate-less group-authenticated key agreement protocol for UAV networks is proposed in [16]. The proposed protocol is based upon elliptic curve cryptography (ECC) and meets the security objectives such as non-repudiation and protection against denial-of-service attacks, but at the cost of significant computational complexity.

A certificate-based proxy signature scheme is recently presented in [17]. The proposed scheme eliminates the need for secure key distribution in identifier-based schemes. However, this scheme is also not lightweight due to the use of public key cryptographic techniques.

A remote user authentication scheme for accessing the data of a UAV drone in real time is proposed by [18]. The proposed scheme for user authentication and key agreement is relatively lightweight and uses a fuzzy extractor for biometric verification of the user. A temporal credential-based anonymous lightweight user authentication scheme for a network of UAVs, called TCALAS is proposed in [19]. A remote user can authenticate itself to a drone in the network for an exchange of data through this proposed scheme. Both [18] and [19] are relatively lightweight authentication schemes and are focused on the external entity that accesses the UAV network. However, an additional issue with the scheme proposed in [19] is its scalability for multiple UAV zones, as identified by [20,21]. Despite some of the abovementioned protocols being lightweight, none of them offer any protection against physical tampering and cloning attacks. The primary reason for this vulnerability in these protocols is due to the storage of authentication data (e.g., challenge/response pairs or secret key) on the non-volatile memory of the UAV, which can be extracted by the adversary through physical or side-channel attacks.

Some recent works have exploited the unclonable property of PUFs for secure authentication of devices in IoT, the Internet of vehicles, mobile ad hoc networks, etc. [22–24]. For example, a secret key generation protocol utilizing SRAM-PUF properties and polar codes is proposed in [25].

### 3. Preliminary Background

A physical unclonable function (PUF) is a physical device that cannot be replicated exactly, and an individual instance of PUF ideally returns a consistent but unpredictable response to every input sequence applied to it [26]. The response or the output of PUF, also referred to as a "hardware fingerprint", to a fixed challenge input sequence, can differ slightly for the same PUF depending upon the ambient conditions, such as moisture, temperature, etc. Because these variations are typically small, they can be removed by error correction techniques and fuzzy extractors. In an ideal PUF implementation, the response to an input sequence for a given instance cannot be predicted from the response generated by other instances. Similarly, the response of the PUF for one input challenge should not aid in predicting its response for any other input sequence. PUFs are one-way functions; i.e., given a response sequence, its corresponding input sequence cannot be predicted.

There are several ways to realize PUFs, but in semiconductor microelectronic devices, the most commonly used PUFs are based upon uncontrollable and unpredictable manufacturing process variations. These variations lend unique delay and ambient characteristics to logic gates, oscillators, and memory cells, which can be exploited to design reliable, consistent, unpredictable, unclonable PUFs. PUFs are very commonly used as a

root of trust for lightweight cryptography protocols [27]. The common classification of PUFs (i.e., weak PUFs and strong PUFs), is made in terms of the cardinality of the input sequence space to which they can reliably respond. The former is limited in terms of its input challenge space containing a unique sequence, whereas the latter type of PUF can reliably respond to a large number of input sequences. Weak PUFs find applications in session key generation protocols, whereas strong PUFs are typically employed for mutual authentication of nodes and users in a network [28]. Recently, researchers have discovered some weaknesses in earlier architectures of the PUF devices that could be exploited to launch machine learning-based attacks. Essentially, these attacks can extract the effect of delays and physical characteristics of the PUF upon its response, and through repeated challenge–response repetition on an instance of the PUF, a machine learning algorithm can efficiently learn the mapping from input (challenge) to output (response) sequences. Some recently proposed PUF architectures exhibit greater robustness to machine learning attacks.

## 4. System Model

In this paper, we consider a multi-agent system composed of a finite number of UAVs connected to a centralized ground station. The UAVs capture monitoring and surveillance data from the environment and can either be static (hovering) or mobile. The captured data is sent to the central station, while the central station manages the network of UAVs by transmitting mission parameters, such as trajectory control and commands. It is assumed that the UAVs also have the capability to communicate and coordinate with each other through permission from the central station. However, the UAVs are assumed to be limited in terms of their onboard resources, such as battery power, processing capabilities, and memory. It is further assumed that the central station is located in a secure and protected location on the ground with an abundance of computing, memory, and energy resources as compared to the individual UAVs. A schematic diagram for the proposed system model is shown in Figure 1. The system model shows legitimate UAV nodes and the ground station alongside adversarial UAV and ground stations. Legitimate aerial nodes (i.e., UAVs) are indicated by $U_i$ and $U_j$, while the ground station is indicated by the symbol $G$. The presence of adversarial nodes $V_\zeta$ in the environment highlights and depicts the threats and vulnerabilities of the network. The adversary may be capable of capturing and tampering with the legitimate devices on the network and may try to authenticate them with the ground station. The proposed solution should be able to detect and isolate such nodes and prohibit them from joining the network.

The objective of this paper is to propose a novel technique for the secure registration and authentication of UAVs with the ground station. The authentication of a pair of UAV devices by the central station can also be followed by the establishment of a secure session between the UAVs. We assume that the UAVs are equipped with a strong PUF device that is capable of generating stable and reliable responses for a multitude of input sequences under a prevalent operational range of ambient conditions. The initial step of authentication (i.e., the registration phase) is carried out prior to the deployment of the UAV stations for the mission. The registration is performed only once before the mission and cannot be repeated during the mission. In the registration phase, a secure channel is assumed between the ground station database and the UAV. A challenge–response pair for the PUF in each UAV station is generated and securely saved in the database of the ground station, but the response is never saved in the UAV device memory. This initial challenge–response pair serves as the root of trust that can be exploited to authenticate the UAV and generate an initial secure session with the ground station after deployment. The assumption of strong PUF leaves open the possibility of generating and sending fresh challenge–response pairs from UAV to the ground station before terminating an already established secure session. In this way, the UAV and ground station can re-authenticate themselves for multiple secure sessions without requiring a new registration phase. In addition, we assume that each registered UAV device has a unique PUF with ideal characteristics embedded inside the processor chip of the UAV such that any attempt to tamper with the chip or access the PUF

will lead to its malfunctioning and render it unusable. In other words, we assume that a physically tampered device will not be able to re-authenticate itself on the network. A list of the assumptions mentioned in this section are also outlined in Table 1.

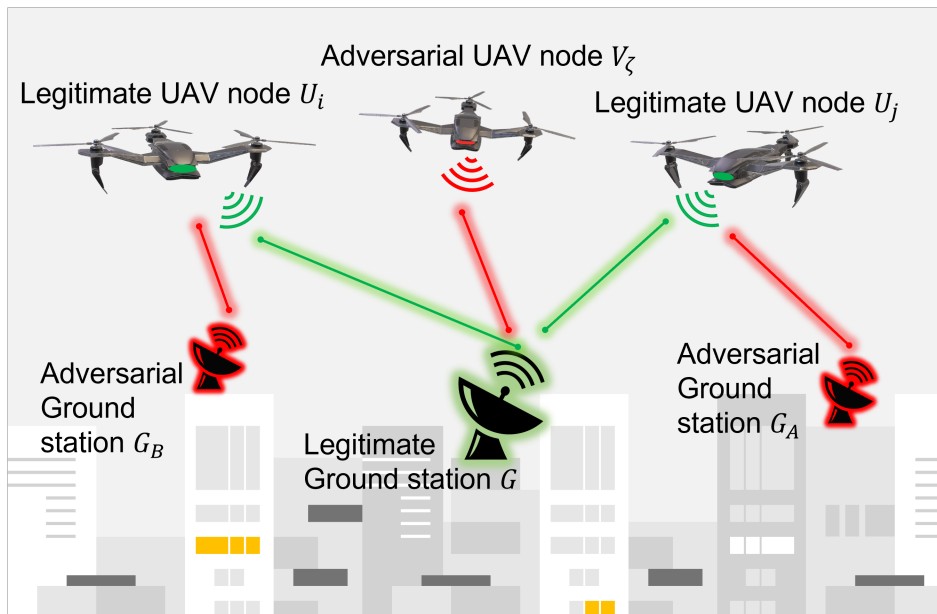

**Figure 1.** The system model shows legitimate UAV nodes and ground station alongside adversarial UAV and ground stations. The ground station is computationally capable as compared to the UAV nodes, which have battery, short-range communication, and limited computing capabilities. Adversarial UAVs may be aided by adversaries on the ground and may be tampered versions of nodes that were captured from the legitimate network. Legitimate UAVs $U_i$ and $U_j$ can authenticate with the ground station $G$, while the authentication mechanism ignores the adversarial devices in the aerial network.

**Table 1.** List of assumptions.

| S. No. | Assumption |
| --- | --- |
| 1. | UAVs have the capability to communicate and coordinate with each other through permission from the central station. |
| 2. | It is assumed that the central station is in a secure and protected location on the ground with an abundance of computing, memory, and energy resources as compared to the individual UAVs. |
| 3. | We assume that the UAVs are equipped with a strong PUF device that can generate stable and reliable responses for a multitude of input sequences under a prevalent operational range of ambient conditions. |
| 4. | The initial step of authentication (i.e., the registration phase) is carried out prior to the deployment of the UAV stations for the mission. |
| 5. | The registration is performed only once before the mission and cannot be repeated during the mission. |
| 6. | In the registration phase, a secure channel is assumed between the ground station database and the UAV. A challenge–response pair for the PUF in each UAV station is generated and securely saved in the database of the ground station, whereas the response is never saved in the UAV device memory. |
| 7. | We assume that each registered UAV device has a unique PUF with ideal characteristics embedded inside the processor chip of the UAV such that any attempt to tamper with the chip or access the PUF will lead to its malfunctioning and render it unusable. |

### 4.1. Threat Models

We assume a strong adversary model, who may actively attempt to masquerade as a legitimate UAV device and initiate various forms of the man-in-the-middle attack, such as packet capturing, modification, replay, and eavesdropping. Due to the wireless nature of the channel and the UAV's operation in the publicly available spectrum band, it is vulnerable to these attacks if secure mutual authentication is not performed. Moreover, the adversary may launch attacks to physically capture the device and attempt to tamper with the device or read its memory contents to create a copy/clone of the device with the same credentials. We refer to the latter type of attack as the cloning attack.

### 4.2. Security Objectives

The protocol proposed in this paper is designed to achieve the following security objectives:

- Secure mutual authentication between the individual UAV devices and the central ground station.
- Resilience to the common attacks such as man-in-the-middle attacks, replay attacks, eavesdropping, etc.
- Generation of secret session key each time a new session is established between a UAV and the ground station.
- Ability to detect and reject captured, tampered, or cloned UAV devices by the adversary.
- Anonymous operation ensured by the use of temporary pseudo-identities for each session established between the UAV and the central station.

## 5. Proposed Authentication Approach

The proposed protocol for secure operation of the network of UAVs can be split into three phases; i.e., the registration phase, authentication of UAV with ground station, and inter-UAV authentication. Table 2 describes the symbols used in each phase of the proposed protocol. Note that a secure session key is generated after mutual authentication between any two devices on the network.

**Table 2.** Symbols and description.

| Symbols | Description |
| --- | --- |
| $U_i$ | Identity of $i$-th UAV |
| $M$ | Number of UAVs |
| $G$ | Identity of the ground station |
| $P_i^{(k)}$ | Pseudo-identity of $i$-th UAV in $k$-th session |
| $C_i^{(k)}$ | $k$-th challenge for PUF in UAV $U_i$ |
| $R_i^{(k)}$ | Response generated by PUF to challenge $C_i^{(k)}$ |
| $\oplus$ | Exclusive-OR operator |
| $\|\|$ | Concatenation operator |
| $\mathcal{M}_n$ | $n$-th authentication message |
| $N_i^{(k)}(n)$ | $n$-th nonce during $U_i$'s $k$-th authentication attempt with $G$ |
| $\mathcal{H}(\mathcal{M})$ | Hash function computed for message $\mathcal{M}$ |
| $S_i^{(k)}$ | Session key for $k$-th session between $U_i$ and $G$ |
| $S_{i,j}^{(k)}$ | Session key for $k$-th session between $U_i$ and $U_j$ |

### 5.1. Registration Phase

The legitimate UAVs $U_i$ are registered with the ground station prior to their deployment for the mission, where $i \in \{0, 1, \ldots, M-1\}$ and $M$ is the number of UAVs on the network. Each UAV is assigned a temporary pseudo-identity $P_i^{(0)}$, which might be refreshed after every secure session. The ground station, however, has a permanent identity $G$. Further steps in the registration phase involve storage of a challenge and response pair

$(C_i^{(0)}, R_i^{(0)})$, as well as the pseudo-identity of the device $P_i^{(0)}$ on both the *i*-th UAV and ground station. The registered UAVs also store the permanent identity $G$ of the ground station. Specifically, $(P_i^{(0)}, C_i^{(0)}, R_i^{(0)})$ is stored in the database of the ground station for each UAV $U_i$, and $(G, P_i^{(0)}, C_i^{(0)})$ is stored in each UAV $U_i$. Note that a UAV $U_i$ does not store the response $R_i^{(0)}$ or pseudo-identities $P_j^{(0)}$ of any UAV $U_j$ other than itself, where $j \neq i$.

*5.2. Authentication Phase*

The second phase of the proposed protocol achieves secure authentication of a UAV device with the ground station. Various messages exchanged in the protocol and the associated computations on the UAV and the ground station are shown in Figure 2. The protocol is designed to ensure that only a valid UAV that has pre-registered with the ground station, can successfully authenticate and start a secure session. A secure session key is also generated as a result of the authentication phase. Various atomic steps involved in this phase are enumerated below:

1. **Initiate Request:** Either a UAV $U_i$ or the ground station $G$ can initiate the request for mutual authentication process by sending a message to the other device. In the case of the ground station initiating the authentication process, the request message is sent along with a message digest, i.e.,

$$\{req, \mathcal{H}(req||G||P_i^{(0)})\}. \tag{1}$$

   For UAV initiating the authentication process, we can proceed directly to the second step.

2. **First Nonce:** The UAV $U_i$ randomly generates a nonce $N_i^0(0)$.

3. **Generate Response from PUF:** The PUF on UAV $U_i$ is utilized to generate response $R_i^{(0)}$ corresponding to the challenge $C_i^{(0)}$ stored in its memory (and in the ground station database).

4. **Compute Hashed MAC at UAV:** A hash is computed $\mathcal{H}(N_i^{(0)}(0), P_i^{(0)}, R_i^{(0)})$ at the UAV $U_i$.

5. **Share Secret with Ground Station:** The UAV $U_i$ sends the nonce $N_i^{(0)}(0)$, its temporary pseudo-identity $P_i^{(0)}$, and the computed hash $\mathcal{H}(N_i^{(0)}(0), P_i^{(0)}, R_i^{(0)})$, i.e.,

$$\mathcal{M}(0) = N_i^{(0)}(0)||P_i^{(0)}||\mathcal{H}(N_i^{(0)}(0), P_i^{(0)}, R_i^{(0)}). \tag{2}$$

   Note that the PUF generated response $R_i^{(0)}$ is not shared directly but only through a message digest.

6. **Secret Verification at Ground Station:** The ground station upon receiving the above message from $U_i$ initiates the verification process:

   (a) To guard against a replay attack, the ground station verifies that the nonce $N_i^0(0)$ received from the UAV $U_i$ is a fresh number never exchanged before between the stations.

   (b) The ground station checks whether the pseudo-identity $P_i^{(0)}$ of the UAV $U_i$ is legitimate or not. This is done by querying its database for the existence of the temporary id $P_i^{(0)}$ of the device.

   (c) The message digest, i.e., $\mathcal{H}(N_i^{(0)}(0), P_i^{(0)}, R_i^{(0)})$ is now verified by the ground station after retrieving the response $R_i^{(0)}$ from its database.

   If any step in the verification by the ground station fails, the request from UAV $U_i$ is deemed to be illegitimate and declined. In this situation, the authentication process is aborted.

7. **Nonce Generation at Ground Station:** After verification, a random nonce $N_i^{(0)}(1)$ is generated at the ground station.

8. **Compose Message from Ground Station:** The response $R_i^{(0)}$, retrieved from the database, can be expressed as the concatenation of two equal-length sequences $R_A$ and $R_B$, i.e.,

$$R_i^{(0)} = R_A || R_B. \tag{3}$$

A message $\mathcal{M}(1)$ with the following contents is then composed at the ground station $G$:

$$\mathcal{M}(1) = Q || \mathcal{H}(N_i^{(0)}(0), N_i^{(0)}(1), Q || G), \tag{4}$$

where $Q$ is given by

$$Q = (A || B) \oplus (R_A || R_B), \tag{5}$$

$$B = R_B \oplus N_i^{(0)}(0), \tag{6}$$

and

$$A = R_A \oplus N_i^{(0)}(1) \oplus B. \tag{7}$$

9. **Response Message from Ground Station:** The message $\mathcal{M}(1)$ is then sent to UAV $U_i$.

10. **Verification at UAV:** Upon receiving $\mathcal{M}(1)$ from the ground station, UAV $U_i$ performs the following computations:

$$(A || B) = Q \oplus (R_A || R_B) \tag{8}$$

$$N_i^{(0)}(1) = A \oplus B \oplus R_A \tag{9}$$

$$N_i^{(0)}(0) = B \oplus R_B. \tag{10}$$

Upon recovering the nonces, $U_i$ verifies the message digest $\mathcal{H}(N_i^{(0)}(0), N_i^{(0)}(1), Q || G)$ by using its record of the ground station identity (saved at the time of registration). If this step is completed successfully, then UAV $U_i$ becomes certain about the integrity of the message and its source. After successful recovery of nonces, the UAV verifies the freshness of the message and that it is generated in response to the message sent by $U_i$. In the event of failure of verification at $U_i$, the authentication process is aborted.

11. **Generating a Challenge for the Next Session:** After verification of the message $\mathcal{M}(1)$, UAV $U_i$ generates another random nonce $N_i^{(0)}(2)$. A part of the new nonce is used as the new challenge $C_i^{(1)}$, that can be used for re-authentication or generating a new session key.

$$C_i = C_i^{(1)} || C_i^{(2)}. \tag{11}$$

The on-chip PUF is used to get the corresponding response, $R_i^{(1)}$ to the new challenge $C_i^{(1)}$.

12. **Encoding the New Response and Nonce:** UAV $U_i$ then generates the following strings:

$$E = R_i^{(0)} \oplus R_i^{(1)}, \tag{12}$$

$$F = N_i^{(0)}(2) \oplus R_A. \tag{13}$$

13. **Generating Session Key at UAV:** The session key for future exchange of information with ground station can now be generated at $U_i$:

$$S_i^{(0)} = (R_A \oplus N_i^{(0)}(1))||(R_B \oplus N_i^{(0)}(2)). \tag{14}$$

14. **Send Message from UAV to GS:** The message $\mathcal{M}(2)$ for the ground station consists of $E, F$ and the message authentication code, i.e.,

$$\mathcal{M}(2) = E||F||\mathcal{H}(R_i^{(1)}||P_i^{(0)}||N_i^{(0)}(1)||N_i^{(0)}(2)||S_i^{(0)}). \tag{15}$$

15. **Recover Nonce and Session Key:** The ground station recovers the new response $R_i^{(1)}$ and the nonce $N_i^{(0)}(2)$, and the secure session key $S_i^{(0)}$ as follows:

$$N_i^{(0)}(2) = F \oplus R_A, \tag{16}$$

$$R_i^{(1)} = E \oplus R_i^{(0)}, \tag{17}$$

and

$$S_i^{(0)} = (R_A \oplus N_i^{(0)}(1))||(R_B \oplus N_i^{(0)}(2)). \tag{18}$$

16. **Verify Message Integrity:** Next, the ground station verifies the message authentication code. If ground station cannot verify successfully, then the authentication is aborted. In the event of successful verification, the new challenge response pair $(C_i^{(1)}, R_i^{(1)})$ for $U_i$ is stored in the secure database of the ground station.

17. **Update the UAV Pseudo-identity**: The pseudo-identity $P_i^{(1)}$ of the device $U_i$ at both the ground station and the UAV are now updated for use in the next session.

$$P_i^{(1)} = \mathcal{H}(R_A||P_i^{(0)}||R_B). \tag{19}$$

This concludes the authentication process between the ground station and an arbitrary UAV on the network. Moreover, a new shared secret in the form of $(C_i^{(1)}, R_i^{(1)})$ is also available to both parties for initiating the second secure session, as needed.

*5.3. Inter-UAV Authentication*

In the next phase, we consider the authentication process between two UAVs $U_i$ and $U_j$, where $0 \le \{i, j\} < M$. Similar to the authentication phase, various messages exchanged in the protocol and the associated communication between the UAVs and the ground station are shown in Figure 3. Without loss of generality, we assume that $i < j$, and that $U_i$ intends to initiate the authentication with the UAV $U_j$. The steps in this phase can be outlined as follows:

1. UAV $U_i$ establishes a secure session with the ground station by following the mutual authentication steps in the previous phase. The session key $S_i^{(0)}$ established as a result of this mutual authentication provides secure communication.
2. The UAV $U_i$ now sends a request to ground station $G$ for establishing a secure session with the UAV $U_j$.
3. The ground station sends an authentication command to the UAV $U_j$, by using its pseudo-identity $P_j^{(0)}$. The command for authentication $auth_j$ is sent along with the message authentication code, i.e.,

$$\{auth_j, \mathcal{H}(auth_j, P_j^{(0)}, G)\}. \tag{20}$$

4. In response, $U_j$ performs the mutual authentication steps (as outlined in the previous phase) with ground station $G$. As a result of this step, a secure session key $S_j^{(0)}$ is now established between the ground station and $U_j$.

5. The ground station, while acting as the central authority, now generates a new key $S_{i,j}^{(0)}$ for the mutual session between $U_i$ and $U_j$.

6. The key $S_{i,j}^{(0)}$ is distributed to $U_i$ and $U_j$ by using symmetric encryption keys $S_i^{(0)}$ and $S_j^{(0)}$, respectively. The two UAVs are now mutually authenticated and can communicate by using the secret key $S_{i,j}^{(0)}$.

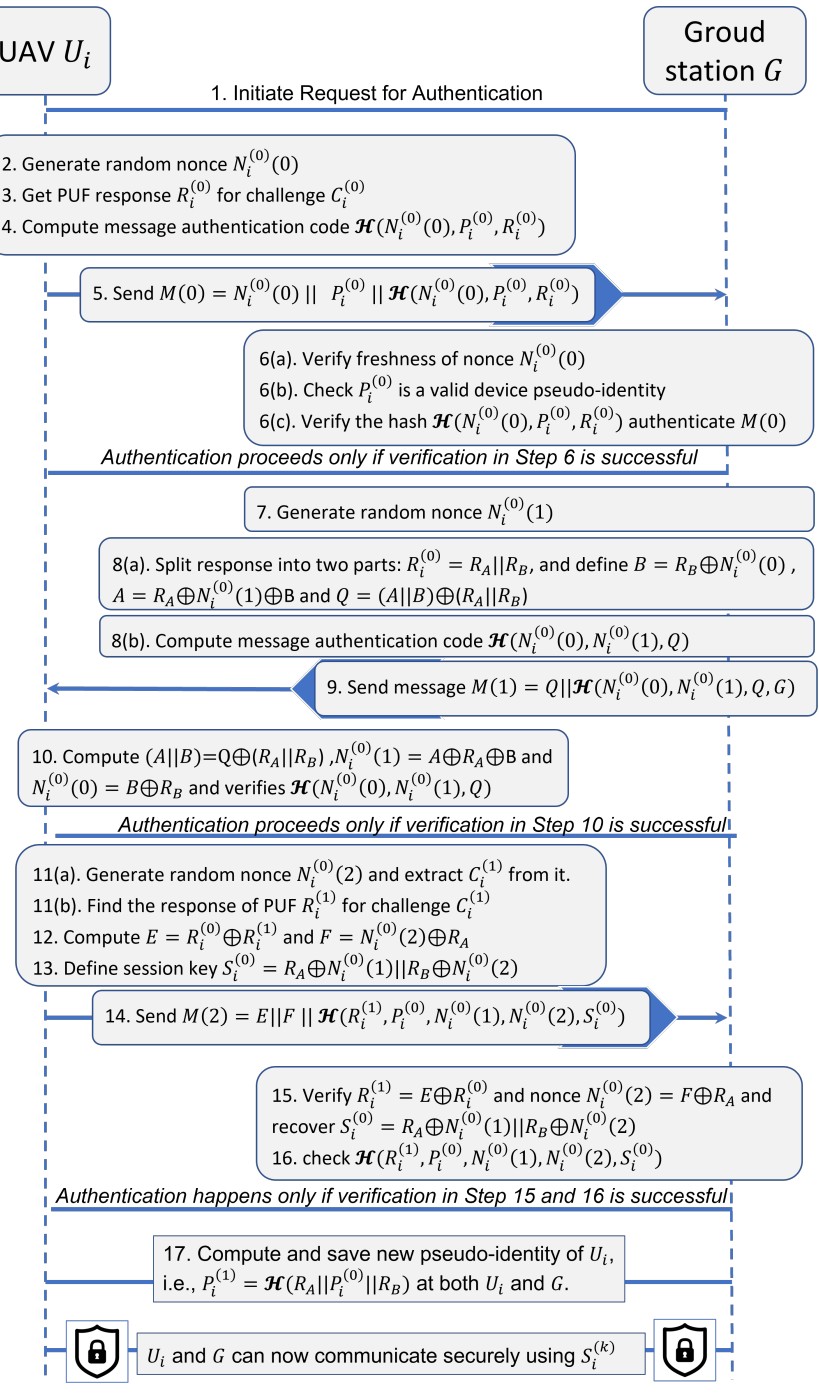

**Figure 2.** UAV authentication steps.

Similar to the above process, any two arbitrary UAVs within the network can mutually authenticate and generate a secret key for pair-wise communication.

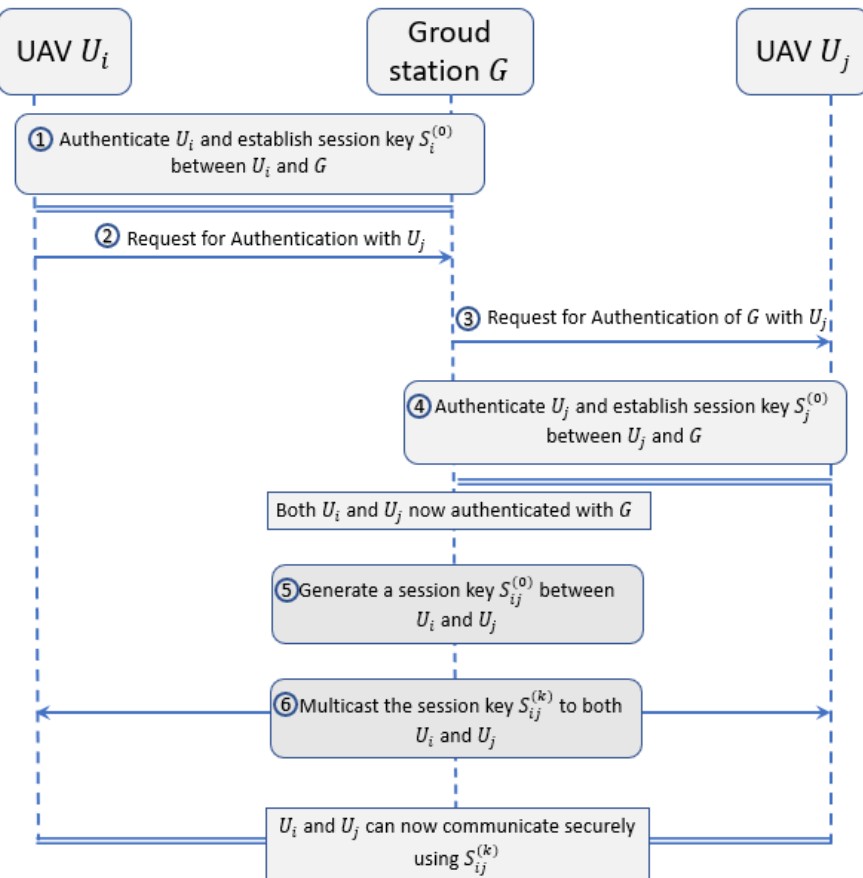

**Figure 3.** Sequence of messages exchanged during inter-UAV authentication. Double lines between devices indicate that the two devices are mutually authenticated and can use a secret key for future secure communication.

## 6. Security Analysis

To assess the security of the proposed protocol for authentication between UAV and ground station, we present security analysis using Mao–Boyd logic [29]. Although we discuss the security analysis only for the UAV to the ground station link, the analysis for inter-UAV authentication is similar as it involves the same steps; these steps are simply repeated twice for each UAV.

*Security Verification by Using Mao–Boyd Logic*

In this proof, we use various rules of Mao–Boyd logic [29], such as nonce-verification, authentication, super-principal, confidentiality, super-principal, intuitive, and group-key inference rules. Symbols related to Mao–Boyd logic and their description are shown in Table 3.

**Table 3.** Symbols for Mao–Boyd logic and their description.

| Logic Symbols | Description |
|---|---|
| $U| \equiv M$ | Principal $U$ believes $M$ is true |
| $U \overset{K}{\leftrightarrow} G$ | The principal $U$ and $G$ share the secret $K$ |
| $U \overset{K}{\lhd} M$ | The principal $U$ has observed the message $M$ through key $K$ |
| $U \equiv \#M$ | The principal $U$ believes that the message $M$ is fresh |
| $G \overset{K}{\not\sim} M$ | $G$ encrypted $M$ using key $K$ |
| $\sup(G)$ | Principal $G$ is the trusted party |

For simplicity of notation, we drop the index $i$ and represent the UAV $U_i$ by $U$ (and similarly for other quantities we drop the subscript and superscripts without sacrificing clarity). Our primary target is to prove the following statement:

**Theorem 1.** *UAV $U$ believes that the nonce $N^{(0)}(1)$ is a good shared secret between the UAV $U$ and ground station $G$.*

**Proof.** Because the response $R$ corresponding to the challenge $C$ is stored in the ground station, so $U$ believes that the response $R$ is a good shared secret between the device $U$ and ground station $G$, i.e.,

$$U| \equiv U \overset{R}{\leftrightarrow} G. \tag{21}$$

By using this shared secret $R$, the UAV is able to compute $Q$ in the return message from the ground station $G$. $Q$ contains both the nonces $N^{(0)}(0)$ and $N^{(0)}(1)$. In other words, $U$ can observe these nonces by using the key $R$, i.e.,

$$U \overset{R}{\lhd} N^{(0)}(0), \tag{22}$$

and

$$U \overset{R}{\lhd} N^{(0)}(1). \tag{23}$$

By using the authentication rule on (21) and (22), we can infer that "U believes $G$ used $R$ to encrypt $N^{(0)}(0)$, i.e.,

$$U| \equiv G \overset{R}{\not\sim} N^{(0)}(0). \tag{24}$$

Similarly, applying the authentication rule on (21) and (23), we can infer that U believes $G$ used $R$ to encrypt $N^{(0)}(1)$, i.e.,

$$U| \equiv G \overset{R}{\not\sim} N^{(0)}(1). \tag{25}$$

$U$ generates a fresh nonce in every round of authentication, so

$$U| \equiv \#N^{(0)}(0). \tag{26}$$

Applying the nonce-verification rule to Equations (24) and (26), we establish that $U$ believes that $G$ believes that $R$ is a good shared secret between $U$ and $G$, i.e.,

$$U| \equiv G| \equiv U \overset{R}{\leftrightarrow} G. \tag{27}$$

Because $G$ generates a fresh nonce in every authentication round, $U$ believes that $G$ believes $U$ has exclusive access to the nonce $N^{(0)}(1)$, i.e.,

$$U| \equiv G| \equiv \{U\}^c \triangleleft ||N^{(0)}(1). \tag{28}$$

The application of the confidentiality rule to (25), (27) and (28) leads to the conclusion that $U$ believes that $G$ believes that only $U$ and $G$ have access to the nonce $N^{(0)}(1)$, i.e.,

$$U| \equiv G| \equiv \{U,G\}^c \triangleleft ||N^{(0)}(1). \tag{29}$$

The proposed authentication mechanism is based upon the assumption that the ground station is trustworthy and secured. This implies that the agent $U$ believes that $G$ is the super-principal, i.e.,

$$U| \equiv sup(G). \tag{30}$$

By using the super-principal rule on (30) and (29), we can conclude that U believes that no one except itself and $G$ has access to $N^{(0)}(1)$, i.e.,

$$U| \equiv \{U,G\}^c \triangleleft ||N^{(0)}(1). \tag{31}$$

From the authentication mechanism described in the previous section, it is apparent that the nonce $N^{(0)}(0)$ is a challenge from $U$, whereas $N^{(0)}(1)$ is the corresponding response from $G$. After receiving the message $\mathcal{M}(1)$ from ground station $G$, the UAV $U$ can decrypt the replied challenge $N^{(0)}(0)$ and response $N^{(0)}(1)$ by using the shared secret $R$,

$$U \overset{R}{\triangleleft} N^{(0)}(0) \; \mathbf{R} \; N^{(0)}(1). \tag{32}$$

The intuitive rule of Mao–Boyd logic implies that U can observe the replied challenge and the response, i.e.,

$$U \triangleleft N^{(0)}(0) \; \mathbf{R} \; N^{(0)}(1). \tag{33}$$

The application of fresh rule on (26) and (33), implies that U believes in the freshness of the nonce $N^{(0)}(1)$, i.e.,

$$U| \equiv \#N^{(0)}(1). \tag{34}$$

By using the good-key rule on (30), (31), and (34), we can readily establish the statement of the theorem, i.e.,

$$U| \equiv U \xleftarrow{N^{(0)}(1)} G. \tag{35}$$

□

Following similar steps, the analogous result for the ground station $G$ can be proved, i.e., $G$ believes that the nonce $N^{(0)}(1)$ is a good shared secret known exclusively to $U$ and $G$. Thus no adversary can access this secret $N^{(0)}(1)$, unless the ground station $G$ itself is compromised. Also, we can show that the nonce $N^{(0)}(2)$ and the new response $R^{(1)}$ are shared exclusive secrets between $U$ and $G$. As a consequence, the secrecy of the session key is ensured, and secure communication between $U$ and $G$ can be established following the authentication phase.

## 7. Comparison

In this section, we present a comparison of the proposed authentication approach with existing schemes in the literature that deal with authentication in a multi-UAV network.

The comparison is summarised in Table 4. The symbols ✓and ✗ represent the presence or absence of a certain security characteristic/protection in the algorithms of [18–20], and the proposed authentication scheme. It can be seen that the proposed scheme represents an attractive choice because it meets all the listed requirements. All the compared algorithms provide forward secrecy, mutual authentication, resilience to man-in-the-middle attacks and replay attacks. Only PUF-based algorithms, such as [20] and the proposed approach provide protection against node tampering and cloning attacks. The PUF-based authentication approaches protect against cloning attacks as the responses are stored in the UAV memory, and hence can never be copied or reproduced by the attacker. Also, an attempt by the adversary to tamper with the PUF device will render the PUF unusable for the attacker. The algorithm in [20] also meets all these specifications except its lack of user anonymity. It does not support anonymity for the users as the user identity is shared on the channel and is not renewed during various authentication phases. Thus, the proposed approach provides secure authentication in the presence of adversaries and removes all vulnerabilities in the face of a realistic threat model consisting of all common attacks.

**Table 4.** Comparison of features for various UAV authentication algorithms.

| Feature | Wazid et al. [18] | Srinivas et al. [19] | Ali et al. [20] | Proposed |
|---|---|---|---|---|
| Clock Synchronization | ✗ | ✗ | ✓ | ✓ |
| User Anonymity | ✓ | ✓ | ✗ | ✓ |
| Forward Secrecy | ✓ | ✓ | ✓ | ✓ |
| Mutual Authentication | ✓ | ✓ | ✓ | ✓ |
| Resilience to Man-in-the-middle (MiTM) Attack | ✓ | ✓ | ✓ | ✓ |
| Resilience to Node Tampering | ✗ | ✗ | ✓ | ✓ |
| Resilience to Replay Attack | ✓ | ✓ | ✓ | ✓ |
| Resilience to Cloning Attack | ✗ | ✗ | ✓ | ✓ |

Now we consider the performance comparison of the proposed algorithm with state-of-the-art UAV authentication schemes in terms of computing requirement, storage and communication overheads. Table 5 shows the comparison between the proposed approach and the authentication schemes in [18–20]. For these computations, we assume that the size of the identity for both the UAV and ground station is equal to 20 bytes. Moreover, we assume that all the schemes utilize a hash algorithm that results in a message authentication code of length 20 bytes. An analysis of the given numbers tell us that the proposed algorithm has the lowest communication overhead and modest storage requirements and hash operations. The number of hash operations, e.g., are less than [18,19] only one more than that of [20]. The number of XOR operations required for the proposed algorithm are more as compared to the competitors, and the proposed algorithm also needs PUF response twice in each authentication round. However, additional operations such as XORs, hashes, and PUF add to the reliability of the authentication approach, and make it more resilient to cloning and tampering attack. Especially, the on board PUF circuitry, which is the root of trust of the proposed authentication algorithm. The PUF onboard UAV devices is impossible to replicate, and it is impossible to predict its response for an arbitrary challenge. These security features are obtained at the cost of very few additional operations as compared to other authentication schemes. The lowest communication and storage overheads make the algorithm lightweight and very attractive for authentication in UAV devices that are energy- and size-constrained.



**Table 5.** Performance comparison of various UAV authentication algorithms.

| Feature | Wazid et al. [18] | Srinivas et al. [19] | Ali et al. [20] | Proposed |
|---|---|---|---|---|
| Computing Cost (Bitwise XORs) | 4 | 3 | 0 | 11 |
| Computing Cost (Random Generators) | 1 | 1 | 5 | 3 |
| Computing Cost (Hash) | 7 | 7 | 2 | 3 |
| Computing Cost (PUF Response) | 0 | 0 | 0 | 2 |
| Storage (Bytes) | 60 | 80 | 40 | 46 |
| Communication Overhead (Bytes) | 212 | 192 | 244 | 128 |

## 8. Conclusions

In this paper, a secure and lightweight authentication scheme is presented for an aerial network of multiple UAVs. The proposed approach derives its root of trust from the physcial, unclonable function that is securely embedded in the memory of UAV nodes. The proposed approach enables secure operation of the UAV network while consuming less on-board resources, such as battery power, computing power, and memory. The security analysis of the proposed approach established that it can provide strong resilience to several cyber and physical attacks, such as man-in-the-middle attacks, replay attacks, masquerading attacks, cloning, tampering, and node capture attacks. Performance analysis shows that the PUF-based authentication algorithm provides a well balanced tradeoff between strong security and resource efficiency.

**Author Contributions:** All authors have substantially contributed towards methodology, validation, formal analysis, and preparation of the original draft. All authors have read and agreed to the published version of the manuscript.

**Funding:** This research received no external funding.

**Data Availability Statement:** Not applicable.

**Conflicts of Interest:** The authors declare no conflict of interest.

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
