# Peer review of "A Lightweight Authentication Scheme for a Network of Unmanned Aerial Vehicles (UAVs) by Using Physical Unclonable Functions"

_electronics, doi:10.3390/electronics11182921_

Round 1

Reviewer 1 Report

In this paper a secure and lightweight authentication scheme is proposed for multiple UAVs in an aerial network. Security analysis were done in section 6 and its very good way described methodologically. Comparisons with past approaches are done in Table 3 and here it is proven for considered many points this work has certain advantages. But elaborate this section and explain comparisons in more details.

Now a very details English grammar checking is needed like in section 7 " the proposed approach provide" it must be provides, this are not expected in research paper. Then terminology are wrong like "main-in-the-middle attack", it is Man in The Middle Attack (MiTM) and all these terminologies should be written properly like Man in The Middle Attack (MiTM)  , first letter capital. So whole paper needs revisions and English grammar checking.  

Reviewer 2 Report

This paper presents a protocol for improving the security of a network of UAVs. 

The equations and most of the mathematical parts look fine. Actually, the paper is enjoyable until the results section, which in my understanding is missing.

This reviewer understands that this kind of contribution usually is hard to present many plots. However, it is necessary to quantify your results in something more than tables.

As the authors mentioned in the work, there are several publications that present results in a good way. So please try to improve your results.

For example, objectives that the authors claim at the beginning of the paper were achieved? How can you show the "Ability to detect and reject captured, tampered or cloned UAV devices by the adversary was achieved"?

Figure 1.- The system model should be made again, there is not a clear relationship between the lines written in the System model section and the figure. The authors just mention Figure 1 in one row. It is important to clarify how is the information moving in the system. Provide a list of assumptions, there are many of them in the System model section, please include all of them in a table.   Figure 2.- Put in two rows the two parts of 8(a).   Step 14.-Send Message from UAV to GS. There is a double "the" in line 323. Education in line 324 should have a number like other similar equations.   Improve the information flow (at least arrows) of Figure 3 and relate each part of the figure with the text. 

Round 2

Reviewer 1 Report

Now as the changes are done so now its a good paper in this domain.

Reviewer 2 Report

Good improvement